# Automated time-height-resolved airmass source attribution for profiling remote sensing applications

Martin Radenz[1], Patric Seifert[1], Holger Baars[1], Athena Augusta Floutsi[1], Zhenping Yin[1,2,3], and Johannes Bühl[1]

[1]Leibniz Institute for Tropospheric Research (TROPOS), Leipzig, Germany
[2]School of Electronic Information, Wuhan University, Wuhan, China
[3]Key Laboratory of Geospace Environment and Geodesy, Ministry of Education, Wuhan, China

**Correspondence:** Martin Radenz (radenz@tropos.de)

**Abstract.** Height resolved airmass source attribution is crucial for the evaluation of profiling ground-based remote sensing observations, especially when using lidar to investigate different aerosol types throughout the atmosphere. Lidar networks, as EARLINET in the frame of ACTRIS, observe profiles of optical aerosol properties almost continuously, but usually additional information is needed to support the characterization of the observed particles. This work presents an approach of how backward trajectories or particle positions from a dispersion model can be combined with geographical information (a land cover classification and manually defined areas) to obtain a continuous and vertically resolved estimate of airmass source above a certain location. Ideally, such an estimate depends on as few as possible a-priori information and auxiliary data. An automated framework for the computation of such an airmass source is presented and two applications are described. Firstly, the airmass source information is used for the interpretation of airmass sources for three case studies with lidar observations from Limassol (Cyprus), Punta Arenas (Chile) and ship-borne off Cabo Verde. Secondly, airmass source statistics are calculated for two multi-week campaigns to assess potential observation biases of lidar-based aerosol statistics. Such an automated approach is a valuable tool for the analysis of short-term campaigns, but also for long-term datasets, for example acquired by EARLINET.

## 1 Introduction

Tracing airmass transport through a turbulent atmosphere is (still) a complex problem. Especially the transport of aerosols and consequently the interactions with clouds, precipitation and radiation are required to capture the four-dimensional history of an air parcel. When it comes to practical application, such as the analysis of aerosol observations or aerosol-cloud interaction studies, the ease of interpretation is often hindered by the amount of data that needs to be considered.

The European Research Infrastructure on Aerosol, Clouds and Trace Gases (ACTRIS) aims at the investigation of short-lived components in the atmosphere, among them aerosols and clouds. As part of ACTRIS, the European Research Lidar Network EARLINET (Pappalardo et al., 2014) operates lidars at more than 25 stations to observe atmospheric state and compositions

up to $30\,\mathrm{km}$ height. The complementary network CLOUDNET (Illingworth et al., 2007) utilizes continuous synergistic observations of ground based instruments such as ceilometers, cloud radars, microwave radiometers and Doppler wind lidars to provide comprehensive cloud observations within Europe and at key regions of the climate system. Both networks, as part
of ACTRIS, need additional, continuous information about airmass source to interpret the observations. Identifying the airmass source region supports the characterization of new particles, e.g. during volcanic eruptions (Pappalardo et al., 2013) or strong wild fires injecting aerosol into the stratosphere (Baars et al., 2019). Also for aerosol typing (e.g. Amiridis et al., 2015; Wandinger, Ulla et al., 2016; Papagiannopoulos et al., 2020; Nicolae et al., 2018; Mylonaki et al., 2020) airmass source can provide an importat constraint. Furthermore, operational height-resolved airmass source information could improve warning
applications for hazardous events, as demonstrated for EARLINET in the frame of the European EUNADICS-AV exercise (Papagiannopoulos et al., 2020).

Models that simulate airmass transport can be broadly grouped into trajectory models and particle dispersion models (overview provided by Fleming et al., 2012). Trajectory models calculate the transport of a single air parcel imposed by the mean meteorological fields. The model simulations can be run either forward or backward in time, providing information
about the source and the destination of the airmass, respectively, after a given transport time. Turbulence and vertical motion during the transport are usually parameterized on the grid scale. Commonly used models are HYSPLIT (Stein et al., 2015), FLEXTRA (Stohl et al., 1995) and LAGRANTO (Wernli and Davies, 1997; Tarasova et al., 2009). Due to the rather simple approach, the results are quite uncertain (Seibert, 1993; Polissar et al., 1999), but computational requirements are comparably low. A straightforward approach to represent some of the variability is to calculate spatial or temporal ensembles of the trajec-
tories (Merrill et al., 1985; Kahl, 1993; Draxler, 2003). Lagrangian particle dispersion models (LPDM) with a large number of particles are set up to cover turbulent and diffusive transport even more realistically (Stohl et al., 2002). The fate of each particle is tracked individually, allowing more variability to be included into the transport simulation. A frequently used LPDM is FLEXPART (Pisso et al., 2019).

Generally, representation of chaotic motion in the atmosphere improves with larger ensembles of trajectories or increasing
number of particles. But, with dozens to hundreds of air parcel locations available, interpretation rapidly becomes cumbersome. A number of infinitesimally small air parcels grouped together gives an airmass, a larger volume of air with similar properties. Residence times are a well established technique for attributing regional information to airmass properties such as being laden with aerosols, moisture or trace gases (Ashbaugh, 1983; Ashbaugh et al., 1985).

Using backward simulations of air parcel positions, analysis of the residence time yields useful information about the poten-
tial source region of an observed airmass. The basic assumption is, that the longer an air parcel was present in a certain region, the more likely it will be influenced by the surface characteristics. Hence, the dimensionality of an air parcels 4D location can be reduced to the residence time. Approaches for clustering backward trajectories by direction, source regions or latitude are widely used. The majority focus on the interpretation of timeseries observations at single heights - mostly close to ground (e.g. Escudero et al., 2011), for aircraft intersects (e.g. Paris et al., 2010) or over a whole region (Lu et al., 2012). More sophisti-
cated approaches blend the residence time with actual concentration measurements (Stohl, 1996; Heintzenberg et al., 2013).

However, these approaches require continuous concentration time series, which are generally not available for remote sensing observations. Furthermore any profile information above the measurement site is neglected.

When interpreting ground-based remote sensing observations, as obtained from aerosol lidars or cloud radars, the airmass sources have been usually assigned by manually selected periods (time and height above ground), that seem interesting for further investigation and calculating backward transport for that specific cases (e.g., Müller et al., 2007; Mattis et al., 2008). If airmass source estimates are required for longer time periods or multiple heights, calculating, visualizing and interpreting the results become tedious. Hence, a continuous, computationally efficient, easy to interpret and automated airmass source estimate is required. To be broadly and easily applicable, such a source estimate should not require extensive a-priori information, such as clusters of trajectories or potential source contribution functions. The required approach is intended to be also simpler than using a coupled aerosol model, such as CAMS (Flemming et al., 2017), COSMO-MUSCAT (Dipu et al., 2017) or ICON-ART (Rieger et al., 2015). Although these models can provide profiles of atmospheric composition, they usually do not provide information on the source.

Herein, we propose a combination of automated backward trajectory calculations and geographical information for the setup of a simple, spatio-temporally resolved airmass source attribution scheme. As a proxy for geographical information, two products are used: a land cover classification mask and manually defined geographical areas. The methodology is described in the following section 2. A comprehensive, easy to use software package is also provided. Earlier versions were already used in Haarig et al. (2017), Foth et al. (2019) and Floutsi et al. (2021). Afterwards, two applications illustrate potential use cases. In the first example, the temporal and vertical evolution of the airmass source is analyzed for three lidar observations of different aerosol conditions from Limassol (Cyprus), Punta Arenas (Chile) and on board R/V Polarstern off Cabo Verde. In the second example, vertically resolved airmass source statistics are used to assess potential observation biases of long-term lidar-based aerosol statistics. Two multi-week campaigns of the PollyNET (Baars et al., 2016), as a part of EARLINET, are presented: Finokalia (Greece) and Krauthausen (Germany).

## 2 Airmass source attribution method

In a conceptualized view, properties of an air parcel arriving over a location of interest are characterized by a certain surface type, if the air was close to the surface during its travelled path. The 'proximity' to the surface can be parameterized as a reception height, which depends on the mixing state of the atmosphere at this location as well as on the type of aerosol particles that could potentially be emitted (i.e. mineral dust or sea salt). Conceivable choices for the reception height are the model-derived depth of the atmospheric boundary layer or fixed thresholds. As a first estimate for identification of possible surface effects on an air parcel, $2\,\mathrm{km}$ is widely used (Val Martin et al., 2018). It is assumed that, the more time an air parcel resides close to the surface, the more likely it will acquire the aerosol footprint of the surface. The residence time - the total time an air parcel spent over a certain surface and below the reception height - is a first hint for the aerosol characteristics of the air parcel.

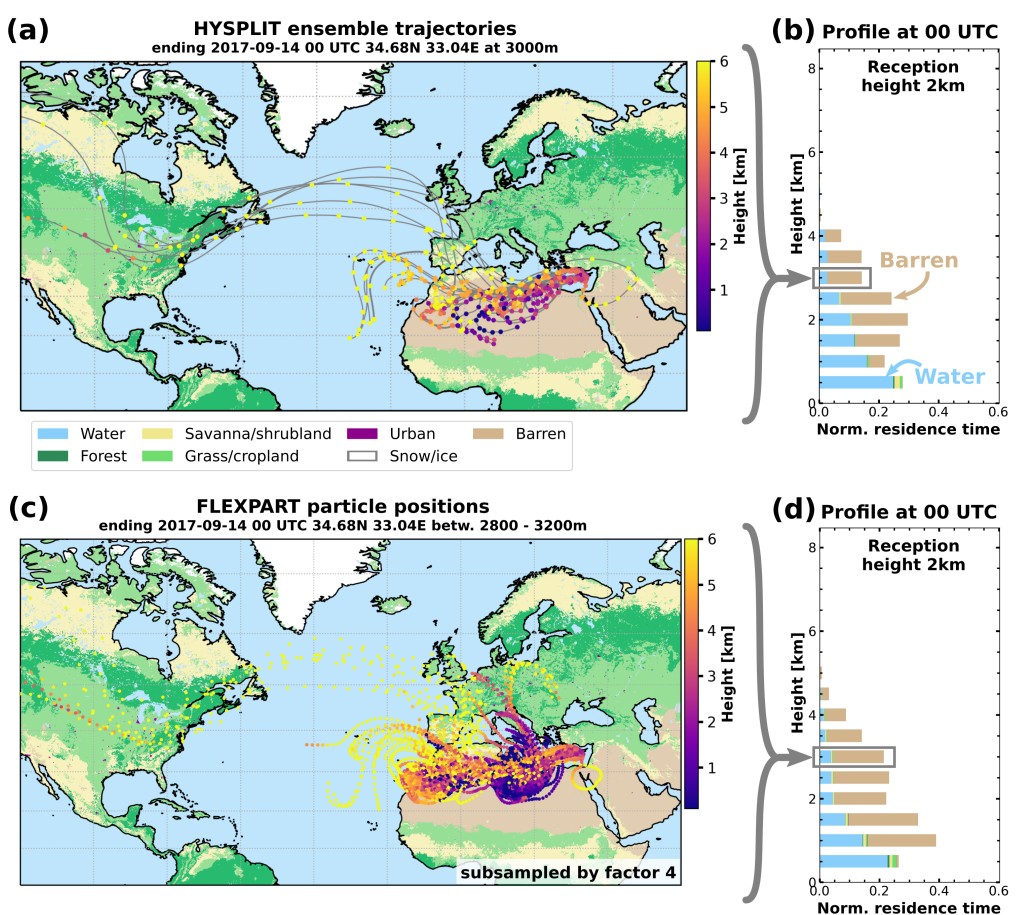

**Figure 1.** Example of how the residence time profile is calculated. HYSPLIT ensemble backward trajectories (a) and FLEXPART particle positions (c) ending above Limassol on the 14 September 2017 00 UTC at 3 km height. The number of FLEXPART particles is reduced by a factor of 4 in this visualization (i.e. 10000 instead of 40000). A time-resolved version with all particles is provided in the supplement. Air parcel height is color-coded. The simplified MODIS land surface classification (Fig. 2) is shown in the background. The profiles of normalized residence time with a reception height threshold of 2 km for HYSPLIT ensemble trajectories (b) and FLEXPART particle positions (d) are shown.

The transport pathway of an airmass arriving over the site can be computed using either mean-wind trajectories or a particle dispersion model. Both approaches can be used with the method proposed in this study. Mean wind trajectories for the past 10 days are calculated using HYSPLIT (Stein et al., 2015). To account for variability, ensemble trajectories consisting of 27 members, spaced 0.3° horizontally and 220 m vertically around the end point, are used (Fig. 1 a). Meteorological input data for HYSPLIT are obtained from the Global Data Assimilation System dataset at 1° horizontal resolution (GDAS1) provided by the Air Resources Laboratory (ARL) of the U.S. National Weather Service's National Centers for Environmental Prediction (ARL Archive). The location of the air parcel is stored in 1 hour steps. A more realistic representation of turbulence and mixing can

90

| MODIS Category | Simplified Category |
|----------------|---------------------|
| 0              | water               |
| 1, 2, 3, 4, 5, 6 | forest            |
| 7, 8, 9        | savanna/shrubland   |
| 10, 11, 12, 14 | grass-, cropland    |
| 13             | urban               |
| 15             | snow                |
| 16             | barren              |

**Table 1.** Overview of how the MODIS land surface categories translate into the simplified categories used in this study. MODIS Category numbers as in (Broxton et al., 2014)

be achieved using a LPDM, which simulates the pathway of hundreds to thousands of particles. Here the most recent version of FLEXPART (Stohl et al., 2005; Pisso et al., 2019) is used. Meteorological data is obtained from the GFS analysis at a horizontal resolution of $1°$ (NOAA, 2000). 500 particles are used with the particle positions being stored every 3 hours. These simulations are run every 3 hours with height steps of $500\,\mathrm{m}$ for the whole period of interest.

In this work, surface is classified by two methods: (1) a simplified version of the MODIS land cover classification (Friedl et al., 2002; Broxton et al., 2014). The 17 categories of the original dataset are grouped into 7 categories according to Tab. 1 in order to allow for robust statistics in the output (Fig. 2). Additionally, the horizontal resolution is reduced to $0.1°$. The categories do not resolve the annual cycles, for example due to growing seasons. (2) customly defined areas as polygons, named according to their geographical context (Fig. 3). These areas can be tailored to the measurement location and/or scientific interest.

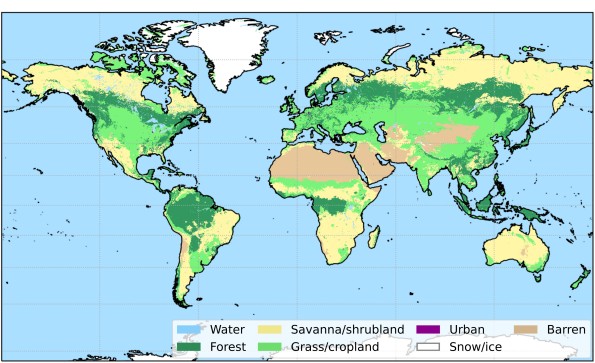

**Figure 2.** The simplified MODIS land cover classification. Details are given in the text.

The residence times at each time and height step are summed for each land cover class or polygon, where the air parcel was below the reception height. Within this study, the widely applicable reception height threshold of $2\,\mathrm{km}$ is used (Val Martin et al., 2018). Different settings can be easily applied to study events which are entrained at greater heights, such as wildfire

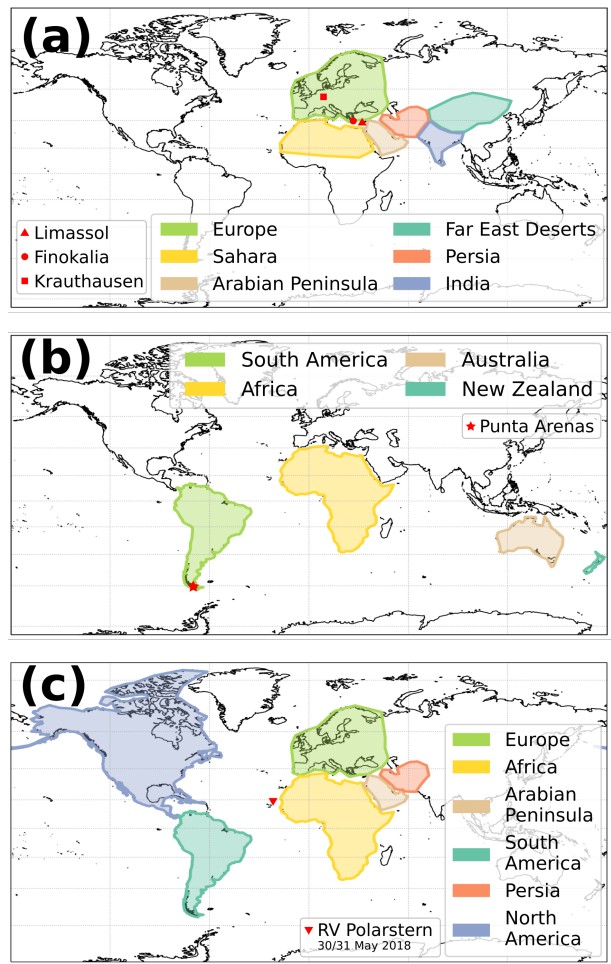

**Figure 3.** The customly defined geographical areas for Limassol, Finokalia, Krauthausen (all a), Punta Arenas (b) and the Atlantic transit (c). Locations of the sites are also marked in the respective map.

smoke emission or volcanic eruptions. The vertical airmass transport during such events is usually not accurately covered by atmospheric models. Setting the reception height to the maximum emission height of such events (as can be estimated, e.g., from satellite observations) can bypass the uncertainties in the modeled vertical motion. The residence times for each category and each height can then be visualized as a profile (Fig. 1 b). Where the residence time is 0, no air parcels were observed below the reception height during the duration of the backward simulation. In the example shown in Fig. 1 (b) above $5\,\text{km}$ height, no airmasses resided at heights below $2\,\text{km}$ above ground in the prior 10 days. The theoretical maximum residence time in hours depends on the number of trajectories or particles $n$, the duration of backward calculation $d$ in days and the interval of output $\Delta o$ in hours:

$$t_{\text{max}} = n\,d\,\frac{24}{\Delta o} \tag{1}$$

To illustrate the temporal evolution, successive airmass source profiles can be shown one after another. This visualization condenses the 4D history of a multitude of trajectories (or thousands of particle positions) to a quickly understandable summary, which structures information on airmass source similar into a time-height cross section. Such a format is usually obtained from vertically or nadir pointed active ground-based remote sensing observations (e.g., Fig. 4).

## 120  3  Polly$^{\mathrm{XT}}$ lidar observations

The airmass source estimate is used to interpret observations conducted with the Polly$^{\mathrm{XT}}$ lidar (Engelmann et al., 2016). Polly$^{\mathrm{XT}}$ is equipped with backscatter-channels at 1064, 532 and 355 nm as well Raman- and depolarization-channels at the shorter two wavelengths. The optical properties are derived using the automated PollyNET retrieval (Baars et al., 2016, 2017; Yin and Baars, 2020) and manual analysis of single profiles. One product of the PollyNET retrieval is the quasi backscatter coefficient, where the attenuated backscatter is corrected for molecular extinction. For this approach, the background, range, and deadtime corrected lidar profiles are normalized by the so called lidar calibration parameter (also sometimes called lidar constant even though it is no constant) which is derived from Raman or Klett retrievals (see Baars et al., 2016). This normalization gives the attenuated backscatter coefficient from ground (note that for the same atmospheric scene, the attenuated backscatter measured from ground is different to the one measured from space, as it is not corrected for attenuation by molecules and particles). The molecular contribution to the atmospheric backscattering and extinction can be calculated from pressure and temperature profiles, the attenuated backscatter coefficient is corrected for the molecular scattering. Furthermore, an assumption of a fixed lidar ratio is applied on the attenuated backscatter corrected for molecular contribution to account for a first guess of the particulate attenuation. This procedure gives the quasi particle backscatter coefficient which is a good proxy for the real particle backscatter coefficient that cannot yet be obtained at high-temporal resolution for all atmospheric scenes. More details are covered in Baars et al. (2017).

Polly$^{\mathrm{XT}}$ was deployed to various field campaigns and longer term measurements during the last 15 years (Baars et al., 2016). A broad variety of meteorological conditions and aerosol regimes was covered. The multi-wavelength observations of Polly$^{\mathrm{XT}}$ contain unique fingerprints of the observed aerosol types from different source regions (Illingworth et al., 2015).

In the following sections 4 and 5, the airmass source attribution will be applied to selected case studies and measurement campaigns, in order to demonstrate its applicability for determination of the airmass source regions and for the estimate of potential observation biases. The case studies are chosen from deployments of Polly$^{\mathrm{XT}}$ to Limassol (Cyprus, 34.7°N, 33.0°E, 12 m a.s.l., October 2016 to March 2018), Punta Arenas (Chile, 53.1°S, 70.9°W, 10 m a.s.l., November 2018 and ongoing) and the RV *Polarstern* Atlantic transit 2018 when passing Cabo Verde (18.1°N, 21.3°W to 21.3°N, 20.8°W). The estimate of potential observation biases is done for two multi-weeks campaigns. One at Krauthausen (Germany, 50.9°N, 6.4°E, 99 m a.s.l.) taking place for 8 weeks in April/May 2013 and the second one at Finokalia (Greece, 35.3°N 25.7°E, 250 m a.s.l.) for 6.5 weeks in June/July 2014.

## 4  Application to lidar case studies

### 4.1  Saharan dust off the coast of West Africa

A lofted layer of dust was observed on 30 and 31 May 2018 by a Polly$^{\mathrm{XT}}$ system on board RV *Polarstern* (Strass, 2018), as the ship steamed between Cabo Verde and African mainland (18.1°N, 21.3°W to 21.3°N, 20.8°W) on her transit north from Punta Arenas (Chile) to Bremerhaven (Germany). A detailed description of the event and optical properties of the observed aerosol were already reported by Yin et al. (2019).

Fig. 4 illustrates the temporal evolution of the observed aerosol plume by means of time-height cross section of the $1064\,\mathrm{nm}$ quasi particle backscatter coefficient for the time period from 30 May 06 UTC to 31 May 06 UTC. Yin et al. (2019) already discussed this case, especially the period from 16 to 17 UTC (their Fig. 14). Optical parameters from the Raman analysis during the following night from 22 to 23 UTC are shown in Fig. 5 (period marked in Fig. 4 (a) with a horizontal orange bar). According to the optical properties Yin et al. argued that the lowest $1\,\mathrm{km}$ was dominated by marine particles and a certain contribution from European continental aerosol. Patchy, liquid clouds were observed at boundary layer top, especially around 09 and 19 UTC. At larger heights, between $1.8$ and $5.2\,\mathrm{km}$ height, a Saharan dust plume with extinction values as large as $700\,\mathrm{Mm}^{-1}$ was present. Lidar ratios were $60\,\mathrm{sr}$ and partlicle linear depolarization ratios at $532\,\mathrm{nm}$ of 0.35. Low Ångström between the lower two wavelengths is consistent with (Veselovskii et al., 2016; Rittmeister et al., 2017). Yin et al. (2019) corroborate their findings by ensemble calculations of HYSPLIT backward trajectories for selected arrival heights and times. However, this way of presentation is rather selective, as information for different heights and times can hardly be shown. This is where the benefit of the continuous airmass source estimate becomes evident. Fig. 6 presents the results of the airmass source estimate for the land surface classification and geographical areas for both, the HYSPLIT (Fig. 6 a,c) and the FLEXPART simulations (Fig. 6, b,d). The estimates based on HYSPLIT and FLEXPART show a good general agreement. The heights and times of certain surface types and geographical regions agree qualitatively. Before 12 UTC on 30 May 2018, FLEXPART derived a lower residence time from barren and grassland or 'Africa', respectively. With respect to Fig. 4, this seems to be reasonable as the layer was rather faint at the beginning of the shown measurement period. Besides this difference, both the HYSPLIT and FLEXPART approaches provide a concise picture of the likely source regions of the observed aerosol. Below $1.5\,\mathrm{km}$ height, the airmass was marine dominated with a small contribution of European grass/cropland. At heights between 2 and $4\,\mathrm{km}$, barren areas from Africa are the main source, but a considerable fraction is also attributed to African grass/cropland and Savanna. This finding is supporting the observations presented by Yin et al. (2019) who already discussed that there was likely a small non-dust fraction in the upper layer, as the particle depolarization ratio profile was not constant at all heights. A potential reason for the observed discrepancy of the observations from pure-dust conditions could be the presence of wildfire smoke stemming from the crop/grassland and savanna. In comparison to the lidar observations, the top of the layer was slightly underestimated by the airmass source estimate. The temporal extent is also fully captured. Variability of backscatter within the layer is not represented by the airmass source estimate, because the strength of dust mobilization is insufficiently parametrized by the reception height. However, the airmass transport is correctly covered by both estimates. Interestingly, the airmass source estimation for this case provides some added value information with respect to the lidar observations. As both HYSPLIT and

FLEXPART approaches indicate, North-American air masses were present in the upper troposphere during the time of the observation, which however had too low aerosol load for being detectable by the Polly$^{XT}$ lidar.

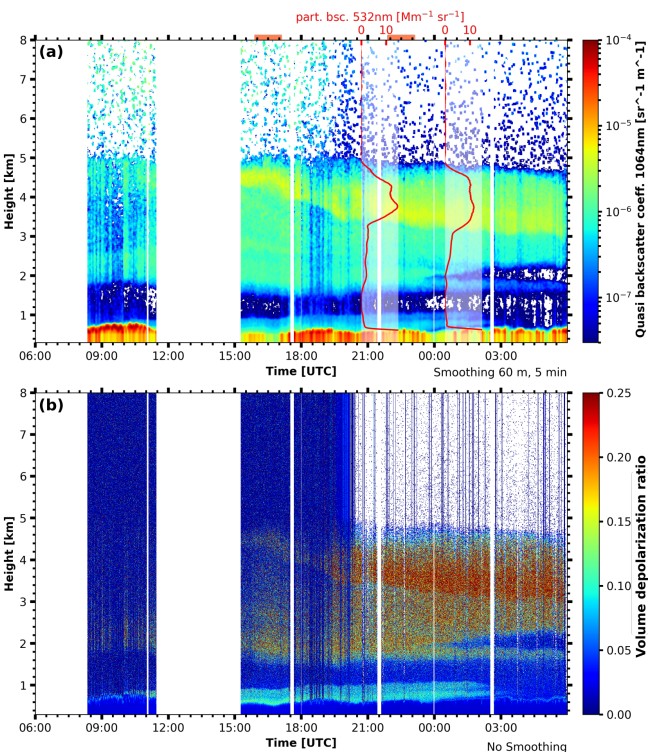

**Figure 4.** (a) Quasi particle backscatter coefficient at $1064\,\mathrm{nm}$ observed by Polly$^{XT}$ on board Polarstern close to Cabo Verde on the 30 and 31 May 2018. Moving average smoothing of 8 range bins ($60\,\mathrm{m}$) and 10 temporal bins (5 minutes) was applied. The red overlays show the Klett derived particle backscatter coefficient from the automated algorithm at $532\,\mathrm{nm}$. The time period of manual analysis (see text) is marked by a horizontal orange bar. (b) Volume depolarization ratio at $532\,\mathrm{nm}$ for the same period. No smoothing was applied.

## 4.2 Saharan and Arabian dust at Limassol, Cyprus

On 14 September 2017 an upper-level short-wave trough moved eastward from the Aegean Sea towards Cyprus. Above $1\,\mathrm{km}$ height, the wind turned from South-West to South during the course of the day with velocities ranging between $5-15\,\mathrm{m\,s^{-1}}$, whereas below, wind velocity was lower and direction more variable.

The time-height cross-section of quasi particle backscatter observed by Polly$^{XT}$ at Limassol shows two pronounced aerosol layers above the boundary layer (Fig. 7). The first layer was observed between $1$ and $2\,\mathrm{km}$ height from 0 to 9 UTC and a second, thicker layer after 3 UTC. Until the night, this layer increases in thickness from bases at 3 and tops at $4.5\,\mathrm{km}$ height to bases at $1.2$ and tops at $6.5\,\mathrm{km}$ height. The boundary layer itself is also laden with aerosols and shows significant backscatter below $1\,\mathrm{km}$ height.

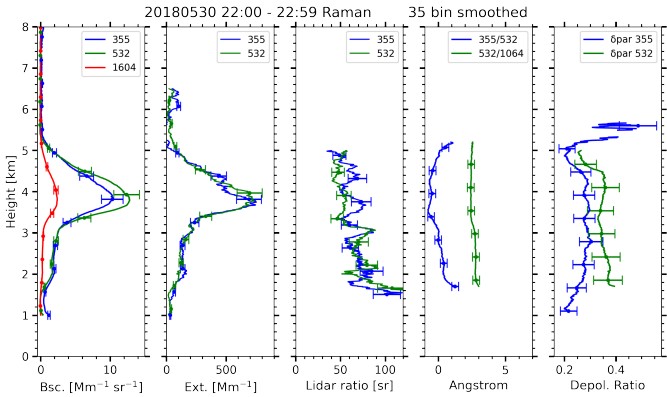

**Figure 5.** Profiles of optical properties on the 30 May 2018 between 22:00 and 22:59 UTC manually derived with the Raman method. A vertical smoothing of 35 bins (262.5 m) was applied.

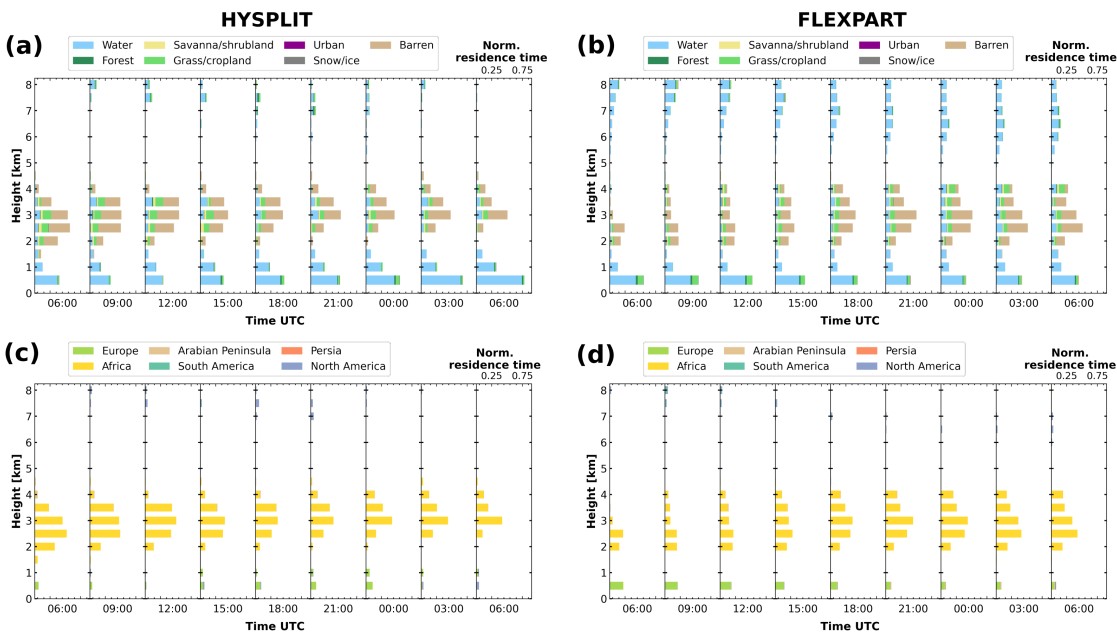

**Figure 6.** Airmass source estimate from 06 UTC on the 30 to 06 UTC on the 31 May 2018 for the land surface classification (a, b) and the named geographical areas (b, d) based on HYSPLIT ensemble trajectories (a, c) and FLEXPART particle positions (b, d).

The optical parameters of the aerosol plume were analyzed for two periods, 02:59 - 04:02 UTC in the morning and 21:41-22:39 UTC in the evening (periods marked in Fig. 7 (a) with horizontal orange bars). The profiles from the morning period (Fig. 8) show for the lower layer at 1.8 km height particle depolarization ratios of 0.25 (355 and 532 nm), low Ångström values and lidar ratios around 40 sr (355 and 532 nm). These optical parameters and their independence of wavelength are typical

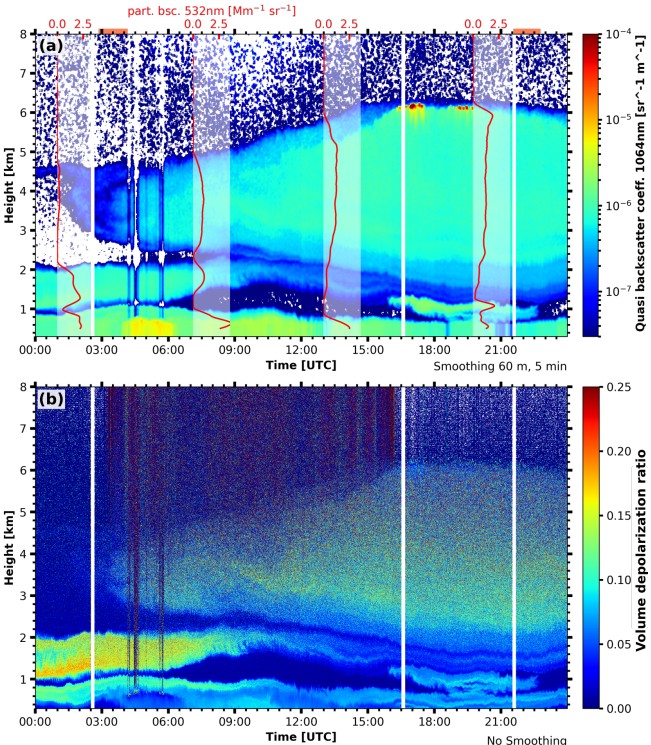

**Figure 7.** (a) Quasi particle backscatter coefficient at $1064\,\mathrm{nm}$ observed by Polly$^{\mathrm{XT}}$ at Limassol on the 14 September 2017. Moving average smoothing of 8 range bins ($60\,\mathrm{m}$) and 10 temporal bins (5 minutes) was applied. The red overlays show the Klett derived particle backscatter coefficient at $532\,\mathrm{nm}$. The time periods of manual analysis (Fig. 8 and 9) are marked by horizontal orange bars. (b) Volume depolarization ratio at $532\,\mathrm{nm}$ for the same period. No smoothing was applied.

for aerosol mixtures with a high dust fraction. Extinction in this layer peaks at $72\,\mathrm{Mm}^{-1}$ (355 and $532\,\mathrm{nm}$). The second layer above $2.5\,\mathrm{km}$ height has particle backscatter values of less than $2\,\mathrm{Mm}^{-1}\,\mathrm{sr}^{-1}$ (at $355\,\mathrm{nm}$) and $0.5\,\mathrm{Mm}^{-1}\,\mathrm{sr}^{-1}$ (at $532\,\mathrm{nm}$). Ångström values are slightly higher than in the lower layer, varying between 1 and 2. The particle depolarization ratios at both, 355 and $532\,\mathrm{nm}$ wavelength, are between $0.05$ and $0.10$. This upper layer during the morning is already the leading edge of the second plume, that increased in thickness during the day (both geometrically and optical). As shown in Fig. 7 (b), the volume depolarization ratio increased only slowly during the averaging period.

During the evening (Fig. 9), the upper layer extended from $1.3$ to $6\,\mathrm{km}$ height and shows homogeneous and mostly wavelength-independent optical properties throughout. Particle depolarization ratios were between $0.10$ and $0.15$, with $532\,\mathrm{nm}$ values slightly higher than at $355\,\mathrm{nm}$. Lidar ratios in that layer were $35\,\mathrm{sr}$, typical for Middle East dust (Mamouri et al., 2013; Nisantzi et al., 2015), while the particle depolarization ratio hints towards a mixture of mineral dust and anthropogenic pollution (e.g. Tesche et al., 2009).

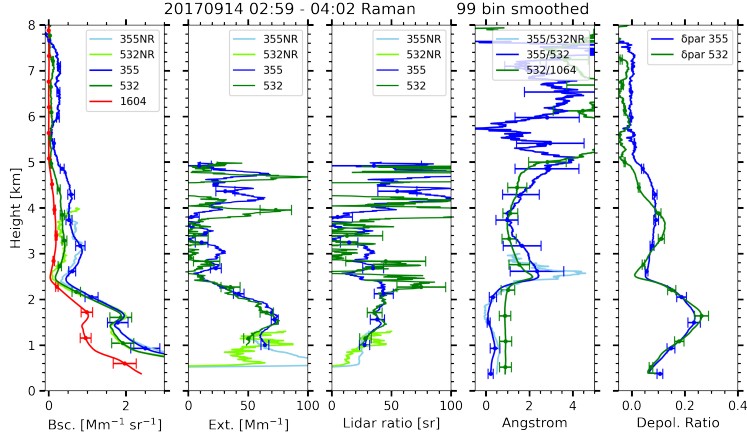

**Figure 8.** Profiles of optical properties on the 14 September 2017 between 02:59 and 04:02 UTC manually derived with the Raman method. A smoothing of 99 range bins (742.5 m) was applied. The abbreviation NR marks profiles observed with the larger field-of-view near-range telescope.

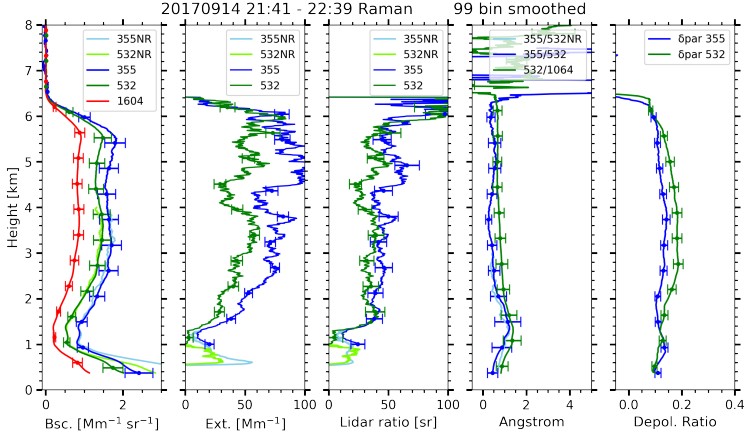

**Figure 9.** Profiles of optical properties on the 14 September 2017 between 21:41 and 22:39 UTC manually derived with the Raman method. A smoothing of 99 range bins (742.5 m) was applied. The abbreviation NR marks profiles observed with the larger field-of-view near-range telescope.

The airmass source estimate (Fig. 10) identifies transport from barren-ground-influenced air from the 'Sahara' until 9 UTC. Later, corresponding to the change in wind direction, the source for the air aloft is identified as 'Arabian Peninsula', but still the barren class. Below 1 km height, a mixture of surfaces was observed, originating mostly form 'Europe'. Comparing the source estimate based on HYSPLIT (Fig. 10 a, c) with the one from FLEXPART (Fig. 10 b, d), both models agree qualitatively well again. While the general transition was captured by the source estimate, the leading edge of the 'Arabian Peninsula' plume was

observed over Limassol earlier than indicated. The increase in thickness of this plume is represented in the source estimate as well.

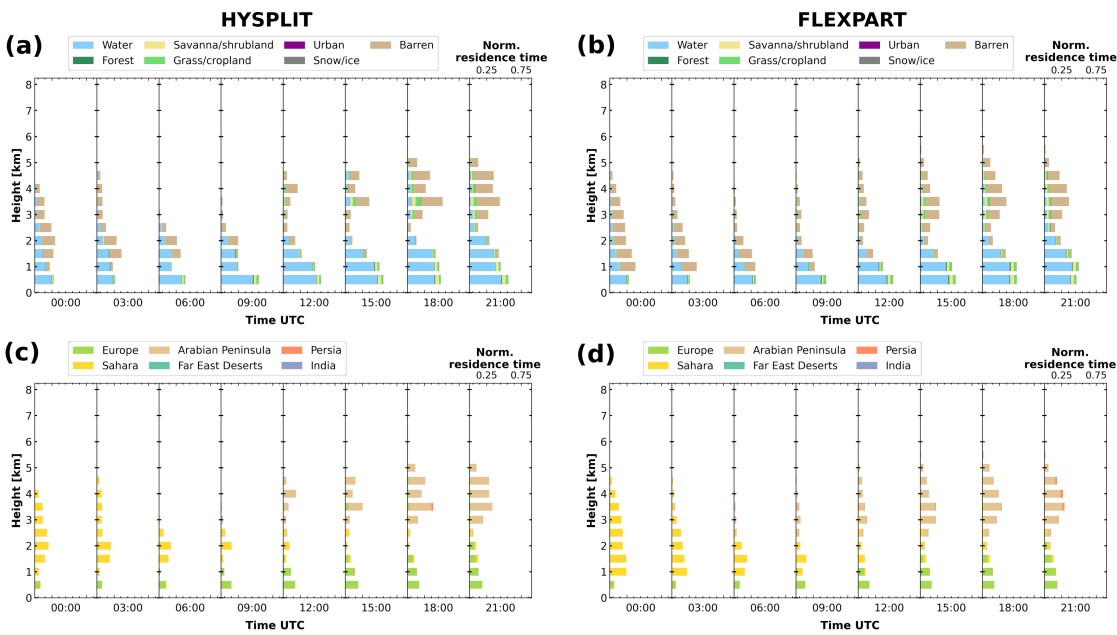

**Figure 10.** Airmass source estimate on the 14 September 2017 for the land surface classification (a, b) and the named geographical areas (b, d) based on HYSPLIT ensemble trajectories (a, c) and FLEXPART particle positions (b, d).

### 4.3 Biomass burning aerosol at Punta Arenas, Chile

Punta Arenas is located in a region where the atmosphere is known to be clean and one of the least affected by anthropogenic influences (Hamilton et al., 2014). Nevertheless, events of aerosol long-range transport occur occasionally (Foth et al., 2019; Floutsi et al., 2021). Due to the large distance of Punta Arenas from aerosol source regions, an attribution of observed aerosol events is in general rather complicated. The application of airmass source estimate for the characterization of an aerosol long-range transport event is presented in here. An upper-level ridge was located off the Chilean coast on 20 May 2019, which

supported also a surface high pressure system. At Punta Arenas the flow was zonal throughout the troposphere. Within that flow long-range transport from across the Pacific Ocean occurred.

In the Polly$^{\text{XT}}$ observations from 20 May 2019 a layer of increased backscatter is present from 2 UTC to roughly 10 UTC. This layer extends from $3\,\text{km}$ to above $6\,\text{km}$ height (Fig. 11). From 14 to 18 UTC a low-level liquid cloud was observed at $1.5\,\text{km}$ height. The cloud was optically thick enough to significantly attenuate the laser beam, causing lack of signal above the

225 clouds top. Occasional cirrus clouds did also enhance the backscatter in the free troppshere, e.g. at 12 UTC between 4 and $5\,\text{km}$. The values of particle backscatter were peaking at $0.3\,\text{Mm}^{-1}\text{sr}^{-1}$ (Fig. 12), which are significantly lower values than reported for the prior cases. In the period analyzed, extinction values were approximately $15\,\text{Mm}^{-1}$ giving lidar ratios well

above $50\,\mathrm{sr}$ and rather low linear particle depolarization ratios. Altogether these optical parameters agree with prior findings of wildfire smoke in the troposphere (Tesche et al., 2011; Burton et al., 2012; Groß et al., 2013; Veselovskii et al., 2015).

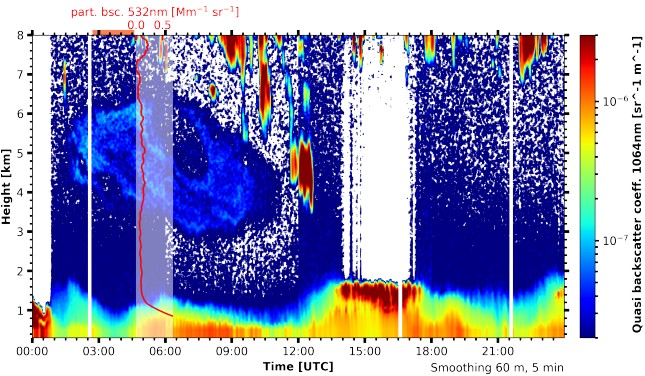

**Figure 11.** Quasi particle backscatter coefficient at $1064\,\mathrm{nm}$ observed by Polly$^{\mathrm{XT}}$ at Punta Arenas on the 20 May 2019. Moving average smoothing of 8 range bins ($60\,\mathrm{m}$) and 10 temporal bins (5 minutes) was applied. The red overlay shows the Klett derived particle backscatter coefficient at $532\,\mathrm{nm}$. The time period of manual analysis (Fig. 12) is marked by a horizontal orange bar.

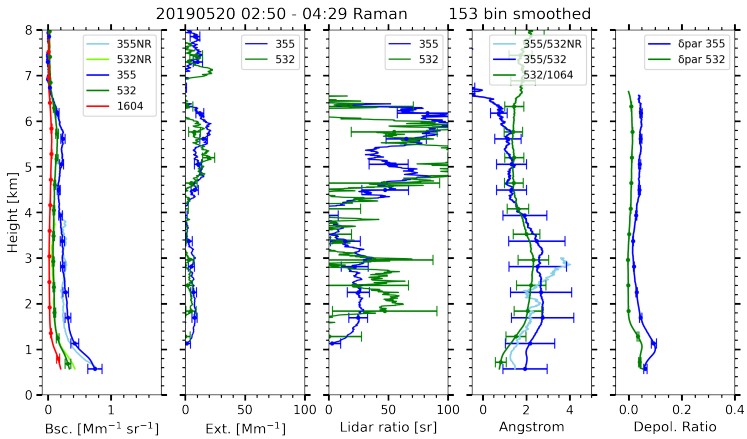

**Figure 12.** Profiles of optical properties on the 20 May 2019 between 02:50 and 04:30 UTC manually derived with the Raman method. A smoothing of range 153 bins ($1147.5\,\mathrm{m}$) was applied. The abbreviation NR marks profiles observed with the larger field-of-view near-range telescope.

The airmass source estimate is also able to capture this faint aerosol layer. Fig. 13 shows, that airmasses form 'Australia' were present between 3 and 9 UTC from 3 to $6\,\mathrm{km}$ height. In terms of land cover class these airmasses were characterized by savanna/shrubland and grass. Wildfires were active in south-western Australia between 10 and 16 May 2019, which is also the region, where the backward simulations end (Fig. A1). Apart from the described period, the airmasses were solely influenced

by the Southern Ocean (i.e. the water class). FLEXPART simulations (Fig. 13 b, d) agree with the HYSPLIT results, however the computed temporal extend and the residence times are slightly longer for the latter. Hence, the airmass source scheme is also capable of capturing aerosol transport at hemispheric (i.e. more than $10000\,\text{km}$) scales.

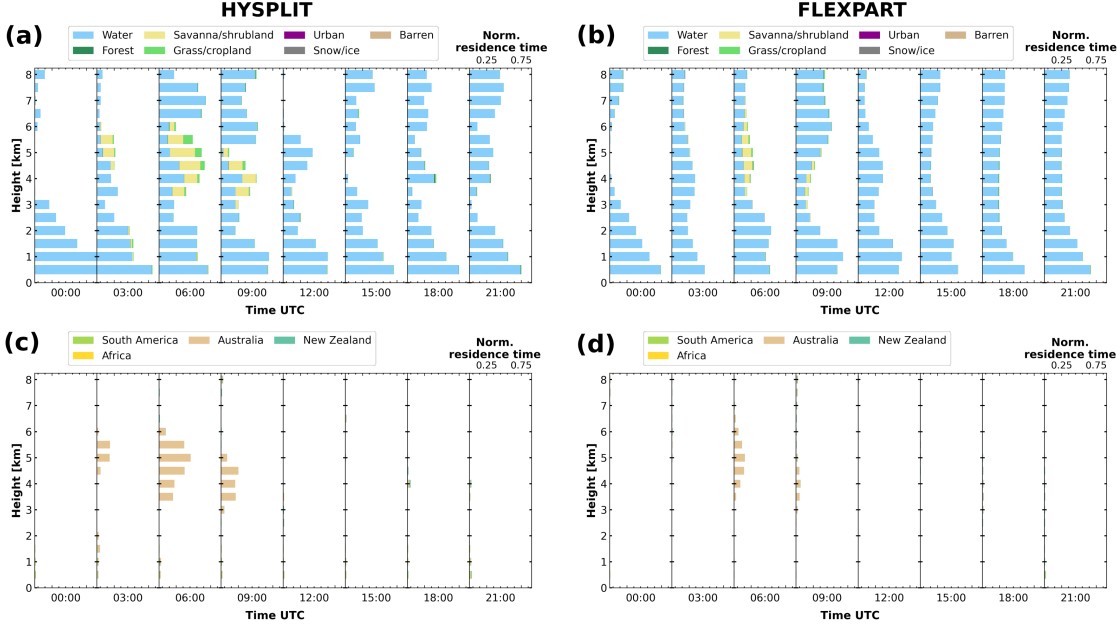

**Figure 13.** Airmass source estimate on the 20 May 2019 for the land surface classification (a, b) and the named geographical areas (b, d) based on HYSPLIT ensemble trajectories (a, c) and FLEXPART particle positions (b, d).

## 5   Assessing potential observation biases

Vertically resolved aerosol statistics are prone to observations biases, as they usually depend on cloud-free conditions. When clouds or precipitation are present, no aerosol properties can be obtained from optical techniques. However, respective statistics, for example, obtained from lidar observations provide key quantities for the determination of the environmental conditions at a certain site (Matthias et al., 2004; Winker et al., 2013; Baars et al., 2016). It is therefore an open question whether the data from suitable (cloud-free) measurement periods are representative for the full observational period. Chances are given that cloudy conditions are related to certain air masses which would stay unidentified in the lidar-based statistics of aerosol optical properties. One way to assess this bias is to compare the airmass residence time statistics of the full observational period with the one subsampled to the times when aerosol information is available.

Applied to lidar data, the automatically analyzed profiles of particle backscatter at $532\,\text{nm}$ from Baars et al. (2016) are used. In their work, the raw profiles are grouped into 30-minute chunks, cloud screened, averaged and analyzed by either the Klett or the Raman method, if signal-to-noise ratio is high enough and a reference height could be set. All profiles that pass a basic

quality control are then included into the backscatter statistics. Obviously, this statistic will only be intermittent, due to overcast
cloud conditions or interruptions in the measurement. Subsampling the airmass source statistics is done by selecting only the
airmass source profiles that are temporally close to a valid lidar profile. A time-threshold of $1.5\,\text{h}$ is used for the following
statistics. However, covering representative airmass conditions is only a necessary condition, not a sufficient one to obtain a
representative aerosol statistics.

Polly$^{\text{XT}}$ observations at Krauthausen (Germany, April/May 2013) and Finokalia (Greece, June/July 2014) are used here. At
Finokalia 940 profiles could be analyzed with the Klett method. Hence, the particle backscatter statistics covers $457.7\,\text{h}$, which
is $42\,\%$ of the campaign duration. The statistics of particle backscatter is shown in Fig. 14 (a). For the Krauthausen deployment
315 profiles could be analyzed with the Klett method, covering $154.2\,\text{h}$ or $11\,\%$ of the campaign. Fig. 15 (a) shows the particle
backscatter statistics.

Profiles of airmass source for the Finokalia deployment are shown in Fig. 14 (b, c). Again with a reception height threshold of
$2\,\text{km}$. The summed residence time of subsampled profiles is divided by the fraction of time covered to make them comparable
to the full residence time. Most dominant land surface categories are water, barren and grass-/cropland. The residence time
of airmasses from barren ground shows a pronounced maximum between 2 and $6\,\text{km}$ height. The residence time of all other
categories decreases monotonically. Airmasses from urban and snow or ice covered areas are 10-100 times less frequent, than
the other categories.

In terms of geographical areas (Fig. 14 c), 'Europe' is the most dominant source up to $3\,\text{km}$ and again above $9\,\text{km}$ height.
Between 3 and $6\,\text{km}$ height the 'Sahara' is the most dominant airmass source. During the campaign period, no airmasses from
the 'Arabian Peninsula', that fulfilled the $< 2\,\text{km}$ criterion were transported to Finokalia.

The dominant sources are well covered by the lidar profiles in terms of land surface, only the barren class is subsampled
by a factor of 10 above $6.5\,\text{km}$ height (Fig. 14 b). This agrees to the Sahara also being subsampled above that height. Air-
masses originating over 'Europe' were also subsampled at heights above $5\,\text{km}$. An undersampling of potentially aerosol laden
airmasses by the lidar statistics will cause the backscatter statistics to be biased low.

During the Krauthausen campaign airmasses originating over water were the most frequent ones, followed by grass-/cropland,
forest, shrubland and barren (Fig. 15 b). Again the residence times of the barren class show a distinct peak between 6 and $8\,\text{km}$
height. Airmasses form the 'Sahara' area agree with the barren class (Fig. 15 c). As expected, 'Europe' is the dominant airmass
source in the lowest $6\,\text{km}$ height, but due to increasing residence times with height for the 'Sahara' source, both are equally
frequent in the upper troposphere. In the lidar observations, 'Europe' is potentially undersampled by $70\,\%$ between 1 and $10\,\text{km}$
height, which is consistent with the grass/cropland and forest class also being undersampled. Barren land surfaces and 'Sahara'
are oversampled by approximately $20\,\%$ up to $7\,\text{km}$ height. In the lowermost $2\,\text{km}$ height the land surface classes urban and
snow/ice also contribute to the airmass mixture and are slightly oversampled.

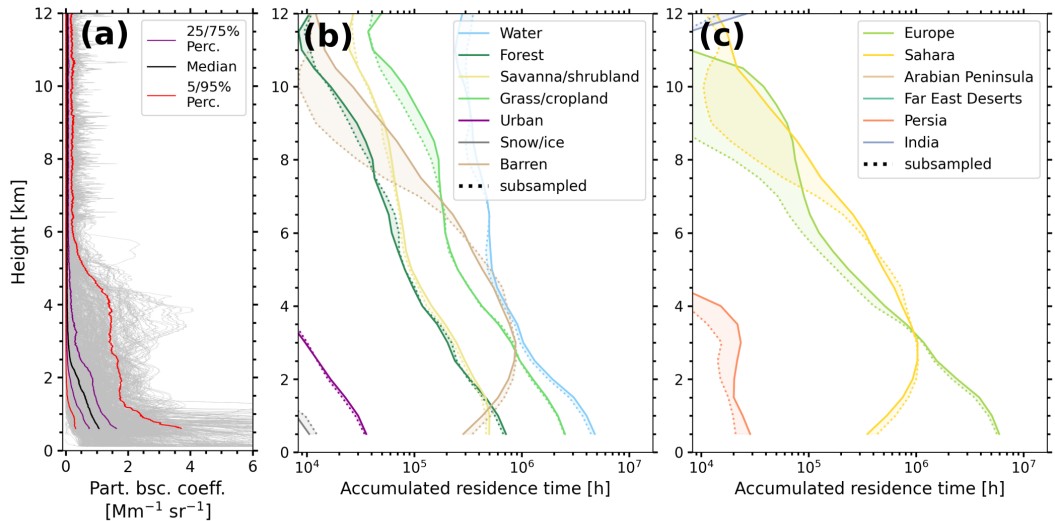

**Figure 14.** Statistics of particle backscatter coefficient (a, as in Baars et al., 2016) and airmass source estimate based on FLEXPART particle positions for the Finokalia campaign of Polly$^{XT}$ in June and July 2014. The land surface classification (b) and the named geographical areas (c) are shown for the full duration (solid) and subsampled only for the periods with available lidar data (dotted). The subsampled residence times are divided by the fraction of time covered. The reception height threshold is 2 km.

## 6 Discussion and Conclusions

In this study we propose an easy to use method for a continuous, height-resolved automated airmass source estimate. By the combination of airmass transport modeling with geographical information, the dimensionality can be reduced and straightforward visualizations accelerate the interpretation of airmass origin. The airmass source estimate can be used to assist (profiling) aerosol observations, as aerosol load and characteristics are strongly controlled by surface properties and atmospheric transport. Three case studies illustrated the applicability at different sites and under different large scale flow conditions In a second application, we showed how the source estimate supports the interpretation of lidar case studies and how potential observation biases can be investigated for longer term campaigns.

The major constraints of the proposed method are discussed in the following. While the airmass transport itself is generally covered well by trajectory models or LPDMs, the linkage to aerosol properties has to be done with care. Firstly, the reception height is modeled by using the mixing depth of the input fields or fixed values for all surfaces and aerosol particles, where differences could be expected for dust, smoke or wildfire smoke. Nevertheless, the assumption for a general reception height might be valid and can be improved in future. The 2 km height used in this work were also reported by other studies (e.g. for wildfires Val Martin et al., 2018) and seem to be applicable over wide ranges of climates and meteorological conditions. Summarizing, a high residence time over a certain class is only a necessary, not a sufficient condition for aerosol load of an air parcel.

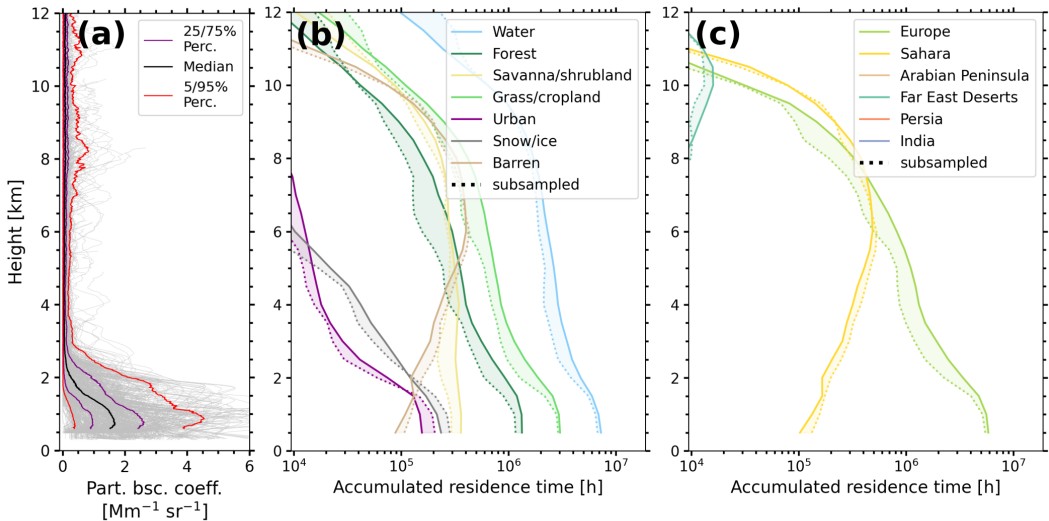

**Figure 15.** Statistics of particle backscatter coefficient (a, as in Baars et al., 2016) and airmass source estimate based on FLEXPART particle positions for the Krauthausen campaign of Polly$^{XT}$ in April and May 2013. The land surface classification (b) and the named geographical areas (c) are shown for the full duration (solid) and subsampled only for the periods with available lidar data (dotted). The subsampled residence times are divided by the fraction of time covered. The reception height threshold is 2 km.

Secondly, aerosol particles might be removed by (wet) deposition between the source and observation site. Currently, such processes are not sufficiently reproduced in trajectory models or LPDMs, as they require detailed representation of aerosol microphysics and precipitation amount. Some improvements in this regard incorporated in the most recent version of FLEXPART (Pisso et al., 2019). However, deposition changes only the aerosol load of an air parcel, not the airmass source itself. Judging from the airmass source residence times alone, this process cannot be distinguished from cases where no emission happened in the first place. These questions could be addressed in future with a full-fledged aerosol transport model that also includes a tracer of airmass origin similar to the scheme shown here.

Some uncertainty is caused by the turbulent nature of the transport. For HYSPLIT a first estimate for the uncertainty of a single parcel location is 20 % of the distance from the trajectories origin (Stohl, 1998). Hence, for HYSPLIT a 27-member ensemble was used, to attribute for this uncertainty. Compared to HYSPLIT, the LPDM FLEXPART allows for a more realistic representation to turbulent transport, as well as a better sampling, when using hundreds or thousands of particles. However, a qualitatively good agreement between the both simulations suggests, that the presented airmass source estimate is rather robust considering uncertainty in the models.

In summary, the described compromises are necessary to get a continuous, height-resolved automated and airmass source estimate. The provided source code allows to use FLEXPART particle positions and HYSPLIT trajectories as an input. User-defined named geographical areas can be easily added. The runtime environment is provided as a docker container, including

FLEXPART v10.4. With that setup one day of airmass source estimate with the resolution used in this study can be processed in less than an hour on a standard desktop computer (2.1 GHz processor, 4 GB RAM, single-threaded).

Such an automated airmass source estimate can provide valuable auxiliary information for the analysis of long-term datasets of profiling aerosol observations, such as collected in the network of EARLINET (Pappalardo et al., 2014). The methodology could also be adapted to existing and future space-borne lidar observations, e.g. CALIPSO (Winker et al., 2009), AEOLUS (Reitebuch, 2012) or EarthCARE (Illingworth et al., 2015). A first estimate of airmass source could be used to constrain retrievals of optical parameters by narrowing the assumed lidar ratio, as in the case of CALIPSO or guide subsequent aerosol typing based on intensive aerosol optical properties, as in the case of AEOLUS and EarthCARE. But, simulating enough air parcels with sufficient along-track resolution might require further development.

With respect to aerosol typing, downstream products such as estimates of concentration of cloud condensation nuclei or ice nucleating particles (Ansmann et al., 2019, 2020) will benefit by the airmass source estimate. Having airmass source information available will advance the implementation of such retrievals into automatic processing, as the single calculus chain (D'Amico et al., 2015) for EARLINET from ground or for EartCARE from space. Also further synergy between lidar target categorizations, such as Baars et al. (2017) and the source estimate remain subject to further investigation.

Apart from the shown applications, the presented methodology can be utilized to assess profiles of airmass source when planning field campaigns. Questions on where, when or how long to measure in order to capture a certain mix of aerosol scenarios can easily be answered. In future the proposed method can be extended by further source maps, for example by dust source maps derived by the approach of Feuerstein and Schepanski (2018) or temporally varying information on wildfires as well as snow and ice cover or biological productivity.

*Code and data availability.* The processing software "trace_airmass_source" as used for this publication is available under Radenz (2021). The most recent version is available via GitHub: https://github.com/martin-rdz/trace_airmass_source (last access: 14.01.2021). A Docker configuration is provied for a straightforward replication of the programming environment, including all dependencies. Meteorological fields for the backward simulations were obtained from ARL Archive and NOAA (2000). The data for the fire radiative power map is available at Giglio (2000). The analysed Polly$^{XT}$ and airmass source data is available on request.

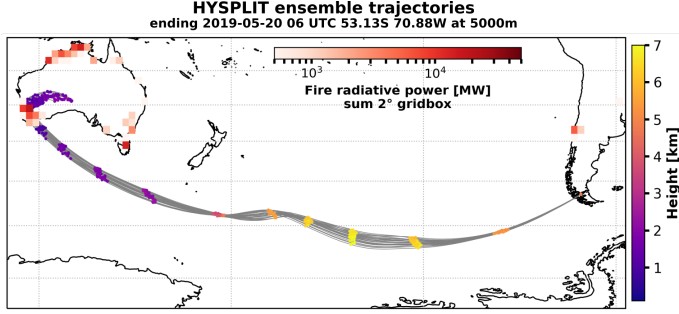

**Figure A1.** HYSPLIT ensemble backward trajectories ending above Punta Arenas on the 20 May 2019 06 UTC at 5 km height together with the MODIS derived fire radiative power (Giglio, 2000). Dots along the trajectories indicate the height of the air parcel in 12 hour intervals. MODIS derived fire radiative power of fires between 10 and 16 May 2019 is gridded to $2°$.

*Author contributions.* MR developed the algorithm and drafted the manuscript. PS, JB supported the implementation and supervised the work. HB, AF and YZ analyzed the lidar data. All authors jointly contributed to the manuscript and the scientific discussion.

*Competing interests.* The authors declare that they have no conflict of interest.

*Acknowledgements.* The research leading to these results has received funding from the European Union's Horizon 2020 research and in-
novation programme under grant agreement no. 654109 (ACTRIS), the European Union Seventh Framework Programme (FP7/2007–2013) under grant agreement no. 603445 (BACCHUS). We acknowledge funding from the Federal Ministry of Education and Research in Germany (BMBF) through the research program "High Definition Clouds and Precipitation for Climate Prediction – HD(CP)2" (grant nos. 01LK1503F,01LK1502I, 01LK1209C, and 01LK1212C). We thank the Alfred Wegener Institute and R/V Polarstern crew and captain for their support (AWI_PS113_00). Many improvements of the Polly instruments, both, in terms of hard and software were triggered by the fruitful discussions and network activities within EARLINET (Pappalardo et al., 2014). We thank Lucia Mona for serving as an editor and gratefully acknowledge the constructive comments by the two anonymous referees.

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
