# Peer review of "Automated time-height-resolved airmass source attribution for profiling remote sensing applications"

_Atmospheric Chemistry and Physics, 2020_

## Referee Comment (RC1) · Anonymous Referee #1 · 11 Nov 2020

The paper by Radenz et al. introduces an effective and user-oriented way to present height-resolved transport modelling simulations associated with aerosol sources. Moreover, the methodology can be applied independently of observations and can be tuned to accommodate unusual aerosol emissions such as volcanic eruptions or intensive biomass burning episodes. The results highlight the robustness and usefulness of the technique when compared against advanced lidar data. Nevertheless, the presentation of the methodology and results should be improved to help the reader understand the ramifications of the study. Moreover, the precision of the language could be improved. There are several issues and technical comments that can improve it. The paper can be published in ACP although it does not fit yet the scope of the

[Figure]

Special Issue (EARLINET aerosol profiling: contributions to atmospheric and climate research). The authors should consider to acknowledge EARLINET if they want to keep the link with this Special Issue given that Polly instruments also participate in the network.

In the following, comments are given for consideration in Specific Comments. The last section lists the Technical Comments.

**Specific Comments**

Ln2 & Ln51: I understand that the distinction "backward trajectories or particle positions" refer to Hysplit and Flexpart respectively. However, in line 51 you refer to "backward trajectories" for the two models. Therefore, I ask you to clarify throughout the text and keep homogeneous wording to avoid confusion.

Ln8 & Ln59: Is it 7- or 8-week campaigns? Please correct.

§1: In the introduction, you describe that trajectory models simulate air parcels while particle dispersion models simulate particles. In my understanding, the terms "airmass" and "air" include the terms above. It would be nice to clarify this in the text.

Ln20–21: Please give the abbreviations and anywhere else.

§2: A summary table with the versions of the models, the meteorological data, the pros and cons, etc. will enhance the clarity of this section.

Figure 1: A similar Figure for Flexpart is needed.

Ln70: What is a "wind trajectory"? I am confused. Hysplit and Flexpart rely on meteorological data to drive the simulations.

Ln83: What does it mean "purpose-serving"? Could you expand on this?

Figure 2 & Figure 3: Both Figures are not introduced and explained in the manuscript!

Figure 3: Figures 3a, 3b, and 3c refer to Limassol, Punta Arenas, and the shipborne observations, respectively. What are the figures for Krauthausen and Finokalia of Section 5? Why the geographical areas are not uniformly defined? Why the Oceans are not included in this selection? What is the reason behind this decision?

Ln94–97: The residence times for Flexpart and Hysplit are not comparable. Could they be normalized?

Pg6Ln106–107: Although Baars et al. (2017) provide the information, a brief description for retrieving the high resolution products should be introduced with the aim to make the manuscript self-contained.

§4.1: It would be nice to see the values of the intensive parameters that characterize the aerosol layers similar to §4.2.

Ln146–149: Could it be that the airmass, although originated from N. America, is aerosol-free over the measuring site? Is it safe to make this assumption? Polly is

a high-performance lidar and it should be used to demonstrate the validity of the methodology. I consider that increasing the averaging either temporally or vertically or both will demonstrate if something resides in higher altitudes.

Figure 4 & Figure 6 & Figure 10: For the sake of completeness, you could report the particle backscatter coefficient for the "orange" sectors? Also, Baars et al. (2017) produce target classification maps that I consider valuable for the assessment of the overall performance of the methodology.

Ln143–144 & Figure 5: What could be the reason that the last 0.5-1 km of the aerosol layer remain undetected? The same is visible for Figure 9.

Ln163–166: What is the origin of this layers? The signature of this layer is somewhat different from a dust layer.

Ln169–170: Is it a mixed dust layer? Is Middle East dust mixed with anthropogenic particles? Can you clarify?

Figure 10: What are the dense backscattering features (e.g., around 4-5 km at 12 utc)? Is it because of the color scale or are they clouds? If they are clouds, shouldn't they be removed?

Figure 12 (a & b): It seems that the profiles indicate "Water" from the ground up to 10 km. This finding is in disagreement with the lidar data. How should we treat this?

Figure 13: Similarly, the simulations indicate significant aerosol transport over 6 km,

whereas the lidar suggests otherwise.

**Technical Comments**

The use of definite and indefinite articles should improved.
Ln2: Add "of" before "how".
Ln6: Remove "exemplary".
Ln12: Replace "trough" with "through". Remove "and entangled".
Ln13: Replace "require" with "are required".
Ln18: Move "either" before "forward".
Ln19: Remove "either" and add "and" in place of "or".
Ln20: Remove "process". Correct to "parameterized".
Ln35: Replace "is" with "are", "done" with "used", and "Most available approaches" with "The majority".
Ln41: Replace "attributed by" with "assigned".
Ln45: Replace "becomes" with "become".
Ln49: Replace "Though" with "Although".
Ln51: Replace "In here" with "Herein".
Ln54: Replace "An" with "A".
Ln58: Remove "application".
Ln59: Remove "out" and "dataset".
Ln63: Replace "past" with "travelled path".
Ln68: Replace "acquires" with "will acquire". Change "spend" with "spent".
Ln69: Add "and" after "surface".
Figure 1: Add "of" after "Example". Rearrange "(a)" and "(b)".
Table 1: Add "of" after "Overview".
Ln77: Add "most" before "recent".
Ln82: Replace "to" to "into".
Ln98: Replace "each other" with "another".

Ln100–101: Please rephrase.

Ln125: Replace "quasi backscatter coefficient" with "quasi particle backscatter coefficient" and everywhere else in the document.

Ln126: Remove "time and height" and "of the observation".

Ln140: Replace "discuss" with "discussed".

Ln145: Remove "only".

Ln151: Remove the first "the".

Ln155: Remove the open parenthesis.

Ln159: Remove "for one period" and "one".

Ln162: Remove the first "of".

Ln167: Remove "height".

Ln168: Remove "it's depth".

Ln169: Replace "middle east" with "Middle East".

Ln180: Remove "also".

Ln182: Remove "at this site".

Ln183: Remove "one".

Ln192: Replace "soruce" with "source".

Ln208: Remove the second "are".

Ln215: Remove "Exemplary, the".

Ln223: Remove "over".

Ln248: Replace "constrains" with "constraints".

Ln251: Remove "for example" and "for a first estimate".

Ln261: Replace "fully-fledged" with "full-fledged".

Ln270: Remove "proved".

Ln274: Replace "approached" with "approach".

---

## Referee Comment (RC2) · Anonymous Referee #2 · 17 Nov 2020

Characterization of atmospheric aerosols using ground based lidar measurements depend upon accurate attribution of the sources of the aerosol. Often this is achieved by selecting specific altitudes and times representing interesting features in the lidar profiles and then running back trajectories from those locations. This paper describes a method to do this in a continuous way by using ensemble trajectories from the HYSPLIT or particle dispersion modelling from FLEXPART. The authors had presented elements of this methodology in earlier publications and have now presented them in a consolidated way. They give examples showing the application of the methodology as well as give an assessment of the representativeness of time limited ground based lidar observations. This is an interesting paper and may have ramifications for other applications.

[Figure]

The basic contents of the manuscript are clear. However the presentation suffers from many typos, grammar and English usage issues. The paper is well within the scope of ACP and will be useful to the remote sensing community. I recommend publication after some revisions. Here are my comments/suggestions in no particular order:

1. While the paper is geared towards the ground based lidar observations, I am wondering if the methodology can be adapted for global spaceborne observations by say CALIPSO and other forthcoming lidars. If the sources of, say the dust layers observed at a remote location by CALIPSO could be reliably and continuously attributed in an automated way, then as a first approximation, one may be able to assign the lidar ratios corresponding to those sources which are known to vary significantly. Similarly, it is conceivable that variable lidar ratios may be assigned to the ageing smoke layers using this method. Using variable lidar ratios in this way should improve the extinction products from elastic lidars like CALIPSO. It will be good if the authors could discuss the feasibility of this scenario, which would add to the value of the paper.

2. I think it will be nice to have validation of some of the results presented. For instance, in Figures 10-12 the authors analyze an aerosol blob between 2-6 km which is estimated to be originating from Australia. From the retrieved lidar ratios and depolarization ratios, it appears that the layer is likely to be lofted smoke. However, it would be good to provide evidence of fire events in Australia around 15-20 May 2019. Do CALIPSO transects close to Punta Arenas on May 20, 2019 show any lofted smoke layer? In Figure 11, The lidar ratio at 532 nm between 2-3 km is about 50 sr and below 1 km is even higher. Would the authors comment on these. There are also some differences between the HYSPLIT and FLEXPART simulations for this case (Figure 12c and 12d). Perhaps a better example would be transported plumes from Australian fire events in January 2020—as described in Ohneiser et al. (2020, ACP, 20, 8003).

3. Why not give the altitude scale in Figures 4, 6 and 10 in km as in the other plots for uniformity? Also the plots for the lidar data and residence time profiles may be shown up to the same altitude for easier comparison.

4. I think some sort of sensitivity study will be useful, e.g running the trajectories for different number of days and checking if this leads to any difference in the results. Similarly, does varying the number of trajectories improve the difference between the HYSPLIT and FLEXPART results discussed (line 135-136) in relation to Figure 5 or Figure 12?

5. Adding location maps showing the points of observation will add context to the Figures.

6. Adding the corresponding depolarization plot for Figure 4 will be helpful.

7. I am a little confused about the features at the lowest altitudes in the lidar observations. For instance in Figure 4, the highest backscatter values occur at the lowest altitudes below 1 km. Firstly, for the sake of completeness, I think it would be better to reproduce the Yin et al. (2019) Figure 14, instead of the reader having to go back to that paper or better still, present the manual analysis for another segment (and include the profile of lidar ratios in that plot). As mentioned in Yin et al. (2019), the extinction coefficients within the MBL are too large to be entirely from the marine aerosols. Could these really be explained by the pollution coming from Europe with their relatively small contribution to the residence times? Similarly, in Figure 10, very high backscatter can be seen between 14-18 UTC around 1.5 km, but the authors do not mention this in the discussion. Is this a measurement artifact? Similar high backscatter blobs can also be seen between 4-8 km at different times in this Figure.

8. Add unit of accumulated residence time in Figure 1b. The accumulated residence times from HYSPLIT and FLEXPART are very different in Figures 5, 9 and 12 and creates confusion for comparison. The reason for this should be clarified in the text.

9. Define NR in legends to the lidar Figures.

10. For completeness it would be good to include an example of the FLEXPART simulations.

---

## Author Comment (AC1) · 14 Jan 2021

**General Remarks**

We thank both reviewers for their time and the constructive comments, which will improve the quality of the paper. The referee comments are formatted in grey and our response in black with indentation and numbering (R##). The line and figure numbers refer to the revised version.

**Specific Reply to Referee #1**

The paper by Radenz et al. introduces an effective and user-oriented way to present height-resolved transport modelling simulations associated with aerosol sources. Moreover, the methodology can be applied independently of observations and can be tuned to accommodate unusual aerosol emissions such as volcanic eruptions or intensive biomass burning episodes. The results highlight the robustness and usefulness of the technique when compared against advanced lidar data. Nevertheless, the presentation of the methodology and results should be improved to help the reader understand the ramifications of the study. Moreover, the precision of the language could be improved. There are several issues and technical comments that can improve it. The paper can be published in ACP although it does not fit yet the scope of the Special Issue (EARLINET aerosol profiling: contributions to atmospheric and climate research). The authors should consider to acknowledge EARLINET if they want to keep the link with this Special Issue given that Polly instruments also participate in the network. In the following, comments are given for consideration in Specific Comments. The last section lists the Technical Comments.

> **R1:** We are now highlighting the contribution of EARLINET to the improvement of the Polly systems in the acknowledgements. Furthermore, we added EARLINET as a potential use case into the discussion.

**Specific Comments**

Ln2 & Ln51: I understand that the distinction "backward trajectories or particle positions" refer to Hysplit and Flexpart respectively. However, in line 51 you refer to "backward trajectories" for the two models. Therefore, I ask you to clarify throughout the text and keep homogeneous wording to avoid confusion.

> **R2:** Thanks for raising that inconsistency. We decided to settle on air parcel position for a single particle position or point in a backward trajectory.

Ln8 & Ln59: Is it 7- or 8-week campaigns? Please correct.

> **R3:** We settled for 'multi-week' in the abstract and introduction. The durations are quantified in Section 3.

§1: In the introduction, you describe that trajectory models simulate air parcels while particle dispersion models simulate particles. In my understanding, the terms "airmass" and "air" include the terms above. It would be nice to clarify this in the text.

> **R4:** From our understanding, an air parcel is an infinitesimally small volume of air, whereas an airmass is a larger volume of air with consistent properties (e.g. composition, origin, aerosol load, moisture content). We added a sentence clarifying this.

Ln20–21: Please give the abbreviations and anywhere else.

> **R5:** As all these abbreviations refer to well-known models and each is cited with a reference, we see no additional benefit of expanding them. We consider this usage covered by the Copernicus house standards:

*"Abbreviations: […] In order to avoid ambiguity, abbreviations that could have numerous meanings must be defined (e.g. "GCM" could stand for "global climate model" or "general circulation model"). This generally does not apply to abbreviations that are better known than their written-out form (e.g. NASA, GPS, GIS, MODIS)."*
We expanded the GDAS1 abbreviation.

§2: A summary table with the versions of the models, the meteorological data, the pros and cons, etc. will enhance the clarity of this section.

**R6:** As meteorological input both models basically use the GFS analysis, however due to technical reasons in two different formats (GDAS as ARL binary and GFS analysis as grib). The FLEXPART version is already mentioned in the text.
We consider a comprehensive discussion of the difference between trajectory models and Lagrangian particle dispersion models beyond the scope of this publication. Please note, that the HYSPLIT model also contains a module for particle simulations, though with a less sophisticated treatment of turbulence.

Figure 1: A similar Figure for Flexpart is needed.

**R7:** We added a respective subfigure to Fig. 1. But please note that only a fraction of the particle positions could be shown. Also suggested by referee #2, please refer to R36.

Ln70: What is a "wind trajectory"? I am confused. Hysplit and Flexpart rely on meteorological data to drive the simulations.

**R8:** The term 'mean wind trajectory' was used to emphasize the non-turbulent nature of the transport calculation, see Stohl (2002).

Ln83: What does it mean "purpose-serving"? Could you expand on this?

**R9:** Too small areas cause fuzzy statistics of residence time. We replaced 'purpose-serving' by 'robust'.

Figure 2 & Figure 3: Both Figures are not introduced and explained in the manuscript!

**R10:** Both references were added.

Figure 3: Figures 3a, 3b, and 3c refer to Limassol, Punta Arenas, and the shipborne observations, respectively. What are the figures for Krauthausen and Finokalia of Section 5?

**R11:** For Finokalia and Krauthausen also Fig. 3 (a) is used. This becomes clear by the categories shown in Fig. 13 and 14 (14 and 15 in revised version). However, we added this explicitly in the text.

Why the geographical areas are not uniformly defined? Why the Oceans are not included in this selection? What is the reason behind this decision?

**R12:** We are not sure what is meant with 'uniformly distributed'. The whole globe is not covered by polygons, because the intent is to specifically test how much residence time can be attributed to the most probable sources. For example, long-range transport across the innertropical convergence zone is rare, hence including southern hemispheric sources to the analysis of the European site would only clutter the statistics, without adding information. Global coverage of the globe is provided by the land surface mask.
Oceans are not included in the geography version, because they are covered with the water surface property of the land surface classification. Aerosol emission by oceans is too transient in it's nature to be attributed to static regions. E.g. it depends of wave

characteristics (height and breaking/non-breaking waves, which is driven by wind speed, ocean currents and seafloor topography) and marine productivity.

Ln94–97: The residence times for Flexpart and Hysplit are not comparable. Could they be normalized?

**R13:** Yes via the formula given in Eq. 1. To avoid further confusion, we decided to normalize the residence time shown in the comparison. Figures 1 b,d, 5, 9 and 12 (revised version 1 b,d, 6, 10, 13) were updated accordingly. Also suggested by referee #2, please see R34.

Pg6Ln106–107: Although Baars et al. (2017) provide the information, a brief description for retrieving the high resolution products should be introduced with the aim to make the manuscript self-contained.

**R14:** We added a description of the quasi particle backscatter to Sec. 3.

§4.1: It would be nice to see the values of the intensive parameters that characterize the aerosol layers similar to §4.2.

**R15:** Was added as new Fig. 5. A similar point was raised by referee #2, please refer to R32.

Ln146–149: Could it be that the airmass, although originated from N. America, is aerosol-free over the measuring site? Is it safe to make this assumption? Polly a high-performance lidar and it should be used to demonstrate the validity of the methodology. I consider that increasing the averaging either temporally or vertically or both will demonstrate if something resides in higher altitudes.

**R16:** Yes, as we state in the manuscript (now lines 278-279), residence time over a certain surface is only a necessary, not a sufficient condition for aerosol load to be present. For example, the wind speeds at the source location could not have be sufficient for dust mobilization or the soil was too moist.

Figure 4 & Figure 6 & Figure 10: For the sake of completeness, you could report the particle backscatter coefficient for the "orange" sectors? Also, Baars et al. (2017) produce target classification maps that I consider valuable for the assessment of the overall performance of the methodology.

**R17:** The orange bars mark the period of the manual analysis. The respective backscatter is shown in the leftmost panel of Fig. 7, 8 and 11 (revised version 8, 9, 12).
We consider including the target classification beyond the scope of this study, as the classification is distinctively different to aerosol typing and would not provide any new information. However, we agree, that the combination of airmass source attribution and the classification or an aerosol typing is an interesting topic for future work. We state this now in the outlook section (now lines 306-207).

Ln143–144 & Figure 5: What could be the reason that the last 0.5-1 km of the aerosol layer remain undetected? The same is visible for Figure 9.

**R18:** (Now lines 160-161 and Fig. 6) In both cases this issue is likely caused by insufficiencies of the meteorological fields used as an input. One could speculate, that the dust load itself changed the dynamics, i.e. by heating the layer and causing lifting. Those effects are not well represented in the analysis used as input for the transport simulations.
For the Punta Arenas case (now Fig. 10), the backscatter plot and the airmass source estimate agree on the top being at 6km.

Ln163–166: What is the origin of this layers? The signature of this layer is somewhat different from a dust layer.

**R19:** The origin of the layers above 2.5 km during the morning period is hard to pinpoint. The airmass source estimate (and thus the model meteorology) suggest barren surface and

Sahara as the sources. But the optical properties (now Fig. 9) and the temporal evolution (now Fig. 7), indicate the leading edge of the Middle East plume (see R20 below). As HYSPLIT and FLEXPART agree quite well, the source for this disagreement is not the transport simulation, but likely the meteorological input fields. This is already noted in the section on the source attribution "While the general transition was captured by the source estimate, the leading edge of the 'Arabian Peninsula' plume was observed over Limassol earlier than indicated." (line 175-176, now lines 195-196).

The time-height plot of depolarization ratio (now Fig. 7b) suggests that the dust fraction of the mixture increased only at 03:30 UTC, causing inhomogeneities over the averaging period. But as the focus of this profile is the lower layer, clearly being attributed to Saharan airmasses we would like to stick to this averaging period. We expanded the section to include this discussion.

Ln169–170: Is it a mixed dust layer? Is Middle East dust mixed with anthropogenic particles? Can you clarify?

**R20:** Yes, we consider this layer being dust mixed with anthropogenic pollution, due particle depolarization ratio, which is lower than would be expected for pure dust. A contribution of absorbing particles of anthropogenic origin is plausible, given industrial areas in the Middle East.

Figure 10: What are the dense backscattering features (e.g., around 4-5 km at 12utc)? Is it because of the color scale or are they clouds? If they are clouds, shouldn't they be removed?

**R21:** (Now Fig. 11). The mentioned features are boundary layer clouds. As Figure 11 is supposed to provide an overview of the measurement scene we decided to keep all features in. We added a sentence to the descriptions (now lines 207-210). See also R33.

Figure 12 (a & b): It seems that the profiles indicate "Water" from the ground up to 10km. This finding is in disagreement with the lidar data. How should we treat this?

**R22:** (Now Fig. 13). Generally open oceans are a weak aerosol source, especially for the free troposphere. We thus do not see an inconsistency here, e.g. Bourgeois et al. (2018, ACP) and Murphy et al. (2019, ACP)

Figure 13: Similarly, the simulations indicate significant aerosol transport over 6 km, whereas the lidar suggests otherwise.

**R23:** (Now Fig. 14). As mentioned in the paper, the lidar undersampled periods with high residence times over barren ground and Europe/Sahara respectively by a factor of up to 10 (now lines 252-255). Such an undersampling happens, if certain transport regimes are correlated with low level cloud cover or other conditions inappropriate for lidar observations, for which no optical properties can be retrieved. We nevertheless added a sentence to clarify that point.

**Technical Comments**

[...]

**R24:** We considered all the technical comments in the revised version of the manuscript, but decided not to quote the list in the response. Thanks for your effort of pointing them out.

**Specific Reply to Referee #2**

Characterization of atmospheric aerosols using ground based lidar measurements de-pend upon accurate attribution of the sources of the aerosol. Often this is achieved by selecting specific altitudes and times representing interesting features in the lidar pro-files and then running back trajectories from those locations. This paper describes a method to do this in a continuous way by using ensemble trajectories from the HYSPLIT or particle dispersion modelling from FLEXPART. The authors had presented elements of this methodology in earlier publications and have now presented them in a consoledated way. They give examples showing the application of the methodology as well as give an assessment of the representativeness of time limited ground based lidar observations. This is an interesting paper and may have ramifications for other applications.

The basic contents of the manuscript are clear. However the presentation suffers from many typos, grammar and English usage issues. The paper is well within the scope of ACP and will be useful to the remote sensing community. I recommend publication after some revisions. Here are my comments/suggestions in no particular order:

**1.** While the paper is geared towards the ground based lidar observations, I am wondering if the methodology can be adapted for global spaceborne observations by say CALIPSO and other forthcoming lidars. If the sources of, say the dust layers observed at a remote location by CALIPSO could be reliably and continuously attributed in an automated way, then as a first approximation, one may be able to assign the lidar ratios corresponding to those sources which are known to vary significantly. Similarly, it is conceivable that variable lidar ratios may be assigned to the ageing smoke layers using this method. Using variable lidar ratios in this way should improve the extinction products from elastic lidars like CALIPSO. It will be good if the authors could discuss the feasibility of this scenario, which would add to the value of the paper.

> **R25:** Thanks for pointing out this aspect. We added a brief paragraph into the discussion (lines 298-303).

**2.** I think it will be nice to have validation of some of the results presented. For in-stance, in Figures 10-12 the authors analyze an aerosol blob between 2-6 km which is estimated to be originating from Australia. From the retrieved lidar ratios and depolarization ratios, it appears that the layer is likely to be lofted smoke. However, it would be good to provide evidence of fire events in Australia around 15-20 May 2019. Do CALIPSO transects close to Punta Arenas on May 20, 2019 show any lofted smokelayer? In Figure 11, The lidar ratio at 532 nm between 2-3 km is about 50 sr and below1 km is even higher. Would the authors comment on these. There are also some differences between the HYSPLIT and FLEXPART simulations for this case (Figure 12cand 12d).

> **R26:** There were various wildfires in southwestern Australia from the 13 to 16 May. A map combining MODIS fire radiative power and an ensemble trajectory was added to the appendix (Fig. A1). Fires in the central part of Chile (Bio Bio province, 1700 km north of Punta Arenas) during the same period were not touched by any trajectory. We checked the CALIPSO overpasses intersecting the backward simulation within 200km and 2 hours from the 15 to the 20 May, but could not find any similar plumes. The areas of interest were frequently covered by cirrus clouds. The high lidar ratio below 1km was caused by a deajustment of the 607 near-range channel. We decided to omit the near-range extinction and lidar ratio in that figure (now Fig. 12).

Perhaps a better example would be transported plumes from Australian fire events in January 2020—as described in Ohneiser et al. (2020, ACP, 20, 8003).

> **R27:** We do not consider the Ohneiser 2020 a useful example to validate our method. Transport processes across the tropopause are not well captured in the used meteorological data and

the model physics. As can be seen in their Fig. 2., the reception height would have to been set to 20km to include the pyroCB lifting. Such a high reception height degrades the selectivity of the residence times and makes those instances harder to interpret.

**3.** Why not give the altitude scale in Figures 4, 6 and 10 in km as in the other plots for uniformity? Also the plots for the lidar data and residence time profiles may be shown up to the same altitude for easier comparison.

> **R28:** Thanks for pointing out this inconsistency; we decided to settle on 8km as top for the plots related to the case studies. The height unit of the quasi particle backscatter plots (now Fig. 4, 7 and 11) is now km.

**4.** I think some sort of sensitivity study will be useful, e.g. running the trajectories for different number of days and checking if this leads to any difference in the results. Similarly, does varying the number of trajectories improve the difference between the HYSPLIT and FLEXPART results discussed (line 135-136) in relation to Figure 5 or Figure 12?

> **R29:** A sensitivity study would require extending the manuscript significantly while, in our opinion, the added information would not justify this expansion. The comparison of FLEXPART to HYSPLIT for each case gives an impression of the variability to be expected from different number of air parcels. A more rigorous sensitivity analysis would require the development of a skill metric and longer-term transport simulations with different settings. Generally, the required number of air parcels is related to the area of the source that should be attributed. To captures point sources, e.g. release of radioactive substances or inert chemical tracers, correctly, a several thousands of air parcels are required. When larger areas are the potential sources, such as dust mobilization, a couple of hundred to a few thousand particles are usually sufficient to capture emission and transport.

**5.** Adding location maps showing the points of observation will add context to the Figures.

> **R30:** We think that adding further figures or subfigures would expand the article too much. Detailed information on the setting of each measurement site is provided in the original literature. However, we added markers to the overview maps in Fig. 3 so that the reader can get an overview in the beginning.

**6.** Adding the corresponding depolarization plot for Figure 4 will be helpful.

> **R31:** The volume depolarization ratio plot was added for the Polarstern and Limassol case (now Fig. 4b and 7b).

**7.** I am a little confused about the features at the lowest altitudes in the lidar observations. For instance in Figure 4, the highest backscatter values occur at the lowest altitudes below 1 km. Firstly, for the sake of completeness, I think it would be better to reproduce the Yin et al. (2019) Figure 14, instead of the reader having to go back to that paper or better still, present the manual analysis for another segment (and include the profile of lidar ratios in that plot).

> **R32:** We added the optical profiles analyzed with the Raman method between 22 and 23 UTC (new Fig 5).

As mentioned in Yin et al. (2019), the extinction coefficients within the MBL are too large to be entirely from the marine aerosols. Could these really be explained by the pollution coming from Europe with their relatively small contribution to the residence times? Similarly, in Figure 10, very high backscatter can be seen between 14-18 UTC around 1.5 km, but the authors do not mention this in the discussion. Is this a measurement artifact? Similar high backscatter blobs can also be seen between 4-8 km at different times in this Figure.

**R33:** (Now Fig 11). In both cases these features are clouds and aerosol particles subject to strong hydroscopic growth. We added a sentence in the description of the case (lines 142-143 and 206-208). Please also refer to R21.

**8.** Add unit of accumulated residence time in Figure 1b. The accumulated residence times from HYSPLIT and FLEXPART are very different in Figures 5, 9 and 12 and creates confusion for comparison. The reason for this should be clarified in the text.

**R34:** This issue is connected to a similar comment by Referee #1, please refer to R13.

**9.** Define NR in legends to the lidar Figures.

**R35:** The abbreviations now described in the figure captions.

**10.** For completeness it would be good to include an example of the FLEXPART simulations.

**R36:** Also suggested by referee #1, please refer to R7.

---

## Author Response (AR2)

**Reply to Review by Editor**

**Editor Decision: Publish subject to minor revisions (review by editor)** (16 Jan 2021) by Lucia Mona
Comments to the Author:
The link to the Special Issue should be furthermore highlighted in the introduction.

**R1:** Thank you for your positive decision. We added a paragraph emphasizing the links to EARLINET/ACTRIS in the introduction. Furthermore, we extended the abstract as well as the discussion respectively.